# Enhancing Neural Theorem Proving
# via High-Quality Data Selection and Verifier Feedback

**Xiaoxue Zhu** [* 1] **Jilin Hu** [* 1] **Fuyuan Zhang** [1 2] **Jianyu Zhang** [1] **Yongwang Zhao** [1 2 3]

## Abstract

Recent advances in large language models have accelerated neural theorem proving (NTP). Isabelle is a mature and important formal theorem prover that has been widely used in software and hardware verification. However, progress in the Isabelle setting remains limited. Existing approaches either optimize search strategies or train on highly imbalanced raw proof corpora. At the same time, the specialized structure of Isabelle proofs limits the effectiveness of general-purpose data selection methods. To address these challenges, we adopt a data-centric framework for neural theorem proving in Isabelle. We characterize high-quality formal proof data along three complementary dimensions—proof complexity, semantic coverage, and reasoning diversity (PSR)—and propose a PSR-guided data selection pipeline to construct a compact, high-quality training subset. In addition, we leverage verifier feedback as a dynamic data signal during inference, introducing a dynamic feedback-based prompt optimization that iteratively incorporates Isabelle verifier feedback to guide proof generation. We construct and release a $4k$ high-quality Isabelle dataset based on the PSR criterion. On the miniF2F-test, fine-tuning solely on PSR-selected data achieves 84.8% Pass@64. When further combined with dynamic feedback–based prompt optimization, the full framework improves performance to 90.6% Pass@64, establishing a new state of the art for neural theorem proving in Isabelle.

---

[*]Equal contribution [1] School of Cyber Science and Technology, College of Computer Science and Technology, Zhejiang University, Hangzhou, Zhejiang, China [2] State Key Laboratory of Blockchain and Data Security, Zhejiang University, Hangzhou, Zhejiang, China [3] Hangzhou High-Tech Zone (Binjiang) Institute of Blockchain and Data Security, Hangzhou, Zhejiang, China . Correspondence to: Yongwang Zhao <zhaoyw@zju.edu.cn>.

*Proceedings of the 43$^{rd}$ International Conference on Machine Learning*, Seoul, South Korea. PMLR 306, 2026. Copyright 2026 by the author(s).

## 1. Introduction

Formal verification provides mathematically rigorous guarantees by encoding system behaviors and mathematical statements in formal logic and discharging them via machine-checkable proofs (Nipkow et al., 2002; Clarke, 1997). However, formal specification and proof construction require substantial expertise and manual effort (Klein et al., 2009), which limits scalability and motivates the development of automated theorem proving (Bibel, 2013; Robinson & Voronkov, 2001). Early approaches relied on symbolic reasoning, with later work incorporating machine learning to assist tasks such as premise selection and proof search (Urban, 2007; Loos et al., 2017).

Recent advances in large language models (LLMs) have further accelerated automated theorem proving by enabling the direct generation of formal proofs in interactive theorem provers (ITPs) such as Lean 4 and Isabelle (Yang et al., 2023; Lample et al., 2022). Under this paradigm, neural theorem proving (NTP) integrates neural generation with symbolic verification to automatically construct formally checkable proofs and reduce manual proof effort.

Lean 4 (Moura & Ullrich, 2021) has emerged as a primary testbed for recent NTP research, offering strong support for large-scale mathematical formalization. Similarly, Isabelle (Nipkow et al., 2002) is a mature and widely adopted interactive theorem-proving system, featuring an expressive logical framework, a structured proof language, and a robust ecosystem of verified libraries. Isabelle has been used to verify real-world software and hardware systems, including seL4 (Klein et al., 2009), verified compilers, and a wide range of formal developments in the Archive of Formal Proofs. Therefore, using large language models to assist Isabelle in such verification tasks is practically meaningful.

Despite these advances, Isabelle-based theorem proving remains far from its full potential. Existing methods primarily rely on supervised training and training-time optimizations (Wang et al., 2024a; Zhao et al., 2024), yet the resulting gains remain limited. Growing evidence suggests that prover performance is constrained more by training data quality than model capacity alone (Pang et al., 2025; Shang et al., 2025). In practice, models are often trained directly

on raw proof corpora (Polu & Sutskever, 2020; Jiang et al., 2022), which exhibit substantial noise and imbalance.

In the broader natural language processing literature, data-centric approaches show that carefully curated small-scale datasets can outperform large unfiltered corpora (Zhou et al., 2023; Chen et al., 2024; Cao et al., 2024). Methods such as MoDS (Du et al., 2023) and self-guided data selection (Li et al., 2024a) further demonstrate that selecting samples based on data quality, coverage, and difficulty can achieve performance comparable to full-data training.

However, formal theorem proving introduces additional constraints that limit the direct transfer of these general-purpose data selection methods (Maric, 2015). The value of a proof sample depends not only on surface-level textual properties but also on its underlying reasoning structure and logical dependencies (Paulson, 1987; Bundy, 1999). Moreover, strict syntactic requirements and verifier-driven feedback in interactive theorem provers necessitate training data that is both formally correct and effective for model–prover interaction (Bertot & Castéran, 2013; Kaliszyk & Urban, 2014).

Motivated by these observations, we characterize high-quality formal proof data along three practical principles:

- **Proof complexity:** preferring proofs with clear structure and moderate difficulty;

- **Semantic coverage:** ensuring broad coverage across theorem domains and problem types;

- **Reasoning diversity:** encouraging diverse proof strategies and reasoning patterns.

We unify these three dimensions into **PSR** (*Proof complexity, Semantic coverage, Reasoning diversity*), a data-centric criterion for scoring and selecting formal proof samples. Based on PSR, we propose a data-centric neural theorem-proving framework for Isabelle that enhances both training and inference by utilizing systematic data. At training time, we perform **High-Quality Data Selection** by scoring proof samples from the raw corpus along the PSR dimensions and selecting a high-quality subset for model fine-tuning. At inference time, prompts and verifier feedback are treated as data signals, and **Dynamic Feedback-Based Prompt Optimization** is applied to iteratively update the model's conditioning based on verifier feedback. Overall, our method proves effective for formal reasoning in Isabelle.

We summarize our main contributions as follows:

- We propose a data-centric neural theorem-proving framework for Isabelle that enhances formal reasoning by unifying PSR-guided *High-Quality Data Selection* at training time with *Dynamic Feedback-Based Prompt Optimization* at inference time.

- We construct a $4k$ high-quality Isabelle training dataset based on the PSR criterion, providing a compact yet effective resource for fine-tuning neural theorem provers and studying data-efficient training. We release the PSR-selected dataset and the fine-tuned LoRA adapter to support future research in formal theorem proving.[1]

- We conduct a comprehensive evaluation on miniF2F-test. Fine-tuning on the high-quality data alone achieves 84.8% Pass@64. When further combined with verifier feedback-based optimization, the full framework improves performance to 90.6% Pass@64, exceeding the previously reported result on Isabelle achieved by ProofAug (Liu et al., 2025).

## 2. Related Work

### 2.1. Neural Theorem Proving

Neural Theorem Proving (NTP) integrates neural models with interactive theorem provers (ITPs) to construct formal proofs. Existing methods can be broadly categorized into proof-step generation and whole-proof generation. Proof-step generation approaches, such as GPT-f (Polu & Sutskever, 2020) and HTPS (Lample et al., 2022), incrementally generate proof steps guided by search and verification signals, and are effective for procedural systems such as Metamath and HOL Light. In contrast, whole-proof generation methods (Jiang et al., 2022) generate complete proof scripts and leverage proof automation to handle intermediate goals, making them suitable for declarative systems such as Isabelle and Lean. More recent work has explored hybrid frameworks that combine neural generation with symbolic guidance or enhanced search, including ProofAug (Liu et al., 2025), Subgoal-XL (Zhao et al., 2024), and LEGO-Prover (Wang et al., 2024b). Despite differences, existing NTP approaches share a common training recipe, relying on supervised training on large proof corpora and emphasizing model design and inference-time reasoning. Our work adopts a data-centric perspective, focusing on how training data quality affects NTP performance in Isabelle.

### 2.2. Data Selection

In the large language model literature, data quality has been shown to play a critical role in model performance. Early studies primarily emphasized scaling training data (Touvron et al., 2023; Yu et al., 2023; Sun et al., 2023), while more recent work has shifted toward data-centric approaches, demonstrating that carefully curated small-scale datasets can substantially improve performance (Cao et al., 2024; Chen et al., 2024; Li et al., 2024b). These findings suggest that data quality, diversity, and structural coverage can often

---

[1]https://github.com/xiaoxuezhu-zju/PSR-Dataset-and-Model

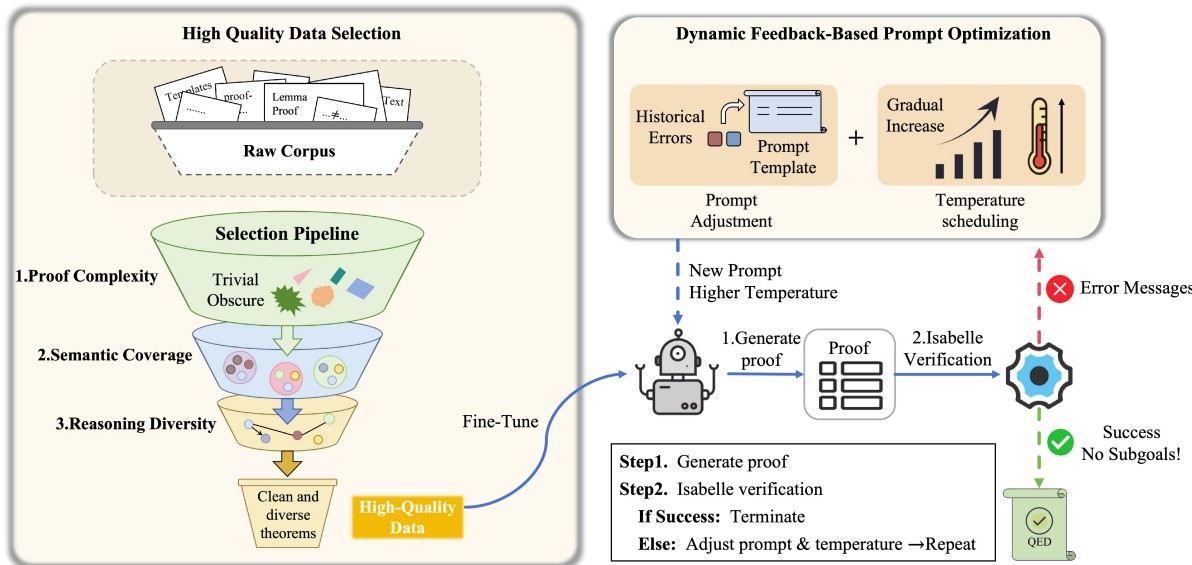

*Figure 1.* **Overview of the data-centric neural theorem-proving framework.** The left panel illustrates *High-Quality Data Selection*, which constructs a curated training subset from the raw proof corpus for model fine-tuning. The right panel depicts *Dynamic Feedback-Based Prompt Optimization*, where the fine-tuned model iteratively generates proof attempts that are verified by Isabelle. If verification succeeds, the process terminates; otherwise, verifier feedback is incorporated to refine subsequent generations.

be more influential than raw data scale alone, particularly in specialized reasoning domains. These methods typically focus on filtering noisy samples, enhancing diversity, or selecting representative examples, and have also been extended to formal theorem proving via large-scale synthetic data generation (Xin et al., 2024). Despite these advances, existing approaches are still largely designed for general language modeling and rely primarily on surface-level textual signals. In contrast, our approach leverages the structural characteristics and formal semantics of Isabelle proofs, tailoring data selection to the specific constraints of interactive theorem proving. This enables the selection process to better reflect the underlying complexity of reasoning and the diversity of proofs inherent in formal verification tasks.

## 3. Methodology

In this section, we present our data-centric neural theorem-proving framework for Isabelle, as illustrated in Figure 1. The framework consists of two main components.

### 3.1. Design Principles

To support effective fine-tuning, we start from an existing corpus (Hu et al., 2025) of approximately 200,000 theorem–proof pairs collected from Isabelle/HOL and the Archive of Formal Proofs (AFP). In the remainder of this paper, the term *raw corpus* refers to this dataset.

Through an empirical analysis of the raw corpus, we identify three recurring data-related issues that limit the effectiveness

of supervised fine-tuning. These observations motivate the three design principles underlying PSR.

**Proof Complexity.** We observe two extremes in the raw corpus:

- one-line or template-based proofs that provide weak structural and reasoning signals;

- extremely long and highly engineered proofs that introduce substantial noise and are difficult for large language models to learn from.

In contrast, proofs with moderate structural complexity exhibit reasoning patterns that are both learnable and transferable. Accordingly, PSR favors such proofs while avoiding the two extremes during selection.

**Semantic Coverage.** The raw proof corpus exhibits substantial imbalance across semantic categories, with some domains heavily overrepresented while others remain sparsely covered. Such distributional skew limits model generalization under a fixed training budget. To address this issue, PSR explicitly promotes broad semantic coverage during data selection. In the Isabelle setting, this entails representing a diverse range of mathematical theories, theorem families, and recurring proof structures, thereby supporting generalization across domains.

**Reasoning Diversity.** Beyond semantic coverage, large portions of the corpus are dominated by a narrow set of proof strategies or tactics, leading to highly homogeneous

reasoning patterns. This redundancy biases models toward brittle behaviors and constrains exploration during inference. PSR therefore prioritizes samples from semantic regions that exhibit greater diversity in proof strategies and tactic usage, explicitly encouraging varied reasoning behaviors to improve robustness and generalization.

## 3.2. Definitions and Metrics

Let the raw proof corpus be denoted as

$$\mathcal{D} = \{(T_i, P_i)\}_{i=1}^N, \tag{1}$$

where $T_i$ denotes a theorem statement and $P_i$ its corresponding proof. Given a data budget $M$, our goal is to construct a compact, information-rich training subset $\mathcal{S} \subset \mathcal{D}$.

### 3.2.1. PROOF COMPLEXITY

We define a composite proof complexity score that combines theorem length, proof length, and explicit reasoning structure markers. Let $\mathcal{T}$ (e.g., `have`, `then`, `show`, `thus`, `proof`, `qed`, etc.) denote a set of structural markers, each corresponding to an explicit reasoning step. The number of such markers in a proof $P_i$ is defined as

$$s_i = \sum_{t \in \mathcal{T}} count(t, P_i), \tag{2}$$

where $count(t, P_i)$ is the number of occurrences of marker $t$ in $P_i$. A key challenge in the raw corpus is the large variance in proof length, where a small number of extremely long proofs can dominate linear metrics. To mitigate this effect and compress the long-tailed distribution, we apply a $\log(1 + \cdot)$ transformation (Schütze et al., 2008) and define the complexity score as

$$c_i = w_p \cdot \log(1 + |P_i|) + w_t \cdot \log(1 + |T_i|) + w_s \cdot s_i, \tag{3}$$

where $(w_p, w_t, w_s)$ are weighting coefficients, and $|P_i|$ and $|T_i|$ denote the textual lengths of the proof and theorem, respectively.

### 3.2.2. SEMANTIC COVERAGE

To ensure that the selected subset $\mathcal{S}$ spans a broad range of mathematical domains, we perform semantic clustering. For each sample, we concatenate the theorem and proof text as

$$t_i = T_i \oplus P_i, \tag{4}$$

and map it to a sparse TF-IDF vector (Schütze et al., 2008)

$$v_i \in \mathbb{R}^d = \text{TFIDF}(t_i; d_{\max}, minDF), \tag{5}$$

where $d_{\max}$ denotes the maximum feature dimension and $minDF$ the minimum document frequency. We then apply K-Means (Lloyd, 1982) to partition the samples into $K$ semantic clusters.

### 3.2.3. REASONING DIVERSITY

We measure reasoning diversity within each semantic cluster using entropy. We define a finite set of strategy labels $A$ (e.g., `linarith`, `ring`, `simp`, `auto`, `smt`, `cases`, `induct`, `rule`, etc.), each corresponding to a commonly used proof tactic in Isabelle.

For each proof $P_i$, we extract a single strategy label $a_i \in A$ using a rule-based matching procedure. Proofs are tokenized using whitespace and punctuation delimiters, and tactic occurrences are identified via exact keyword matching with word-boundary constraints. Strategy labels are matched according to a fixed, predefined priority order over $A$, and the first matched label is assigned to $P_i$. If no strategy label is matched, the proof is assigned to an `other` category. For a semantic cluster $\mathcal{C}_k$, we compute the strategy counts as

$$m_{k,a} = \sum_{x_i \in \mathcal{C}_k} \mathbf{1}[a_i = a], \tag{6}$$

and the corresponding empirical distribution as

$$\hat{P}_k(a) = \frac{m_{k,a}}{\sum_{a'} m_{k,a'}}. \tag{7}$$

Based on Shannon entropy (Shannon, 1948), we define the strategy entropy of cluster $\mathcal{C}_k$ as

$$H_k = -\sum_{a \in A} \hat{P}_k(a) \cdot \log(\hat{P}_k(a) + \epsilon), \tag{8}$$

where $\epsilon$ is a numerical stability term. Higher entropy indicates greater diversity in reasoning strategies, while lower entropy reflects dominance by a small number of tactics. Accordingly, we bias the sampling process toward clusters with higher strategy entropy. Although the absolute entropy values depend on the matching priority, this does not affect the overall data selection behavior in practice.

## 3.3. High-Quality Data Selection

Based on the above definitions, we design a multi-stage data selection pipeline to construct high-quality subsets from the raw corpus.

### 3.3.1. COMPLEXITY ESTIMATION

We first compute a complexity score $c_i$ for each sample $(T_i, P_i)$ in the raw dataset $\mathcal{D}$ according to Eq. (3). To remove extreme samples at both ends of the complexity distribution, we adopt a quantile-based pruning strategy. Given a retention ratio $r$, we compute the lower and upper quantiles $q_{\text{low}}$ and $q_{\text{high}}$ over the complexity distribution of $\mathcal{D}$, and retain the intermediate subset

$$\mathcal{D}_{mid} = \{(T_i, P_i) \in \mathcal{D} : q_{low} \le c_i \le q_{high}\}. \tag{9}$$

---

**Prompt Template Construction (Round $r$)**

**[System Instruction]** You are an expert in Isabelle/HOL theorem proving. Your goal is to prove the given theorem using valid Isabelle theory.

**[Theorem Context]**
Theorem:  theorem mathd_numbertheory_342:
      "54 mod 6 = (0::nat)"

**[Dynamic Feedback History]** (Only if $r > 0$)
**Previous Attempt Analysis:** You have tried to prove this theorem in Round $r$ - 1 but failed.
- Error:Failed to apply proof method
- Failed Response:by (rule mod_mult2_eq [THEN sym]) simp
**Constraint:** Do NOT repeat the exact same invalid step. **Please adjust your response based on the type error above.**

**[Task Trigger]** Please generate the complete proof.

---

*Figure 2.* **Template for Dynamic Feedback-Based Prompt Optimization (round $r$).** The prompt is composed of the problem statement, a summary of prior attempts, and verifier feedback returned by Isabelle.

### 3.3.2. SEMANTIC CLUSTERING

After complexity-based filtering, we perform semantic modeling on the intermediate subset $\mathcal{D}_{\mathrm{mid}}$. Each sample is mapped to a semantic embedding $v_i$, and K-Means is applied to partition the samples into $K$ semantic clusters:

$$z_i = \mathrm{KMeans}(v_i, K) \in \{0, \dots, K-1\}. \quad (10)$$

We denote the set of samples in the $k$-th cluster as

$$\mathcal{C}_k = \{x_i \in \mathcal{D}_{mid} | z_i = k\}, n_k = |\mathcal{C}_k|. \quad (11)$$

### 3.3.3. DIVERSITY MEASUREMENT

To quantify reasoning diversity across semantic clusters, we compute the intra-cluster strategy entropy $H_k$ for each cluster as defined in Eq. (8), and apply min–max normalization:

$$\tilde{H}_k = \frac{H_k - \min_j H_j}{\max_j H_j - \min_j H_j}. \quad (12)$$

To prevent low-diversity clusters from dominating the sampling process, we introduce an entropy-based regularization and define the cluster weight as

$$w_k = n_k \cdot \left(1 - \lambda \cdot (1 - \tilde{H}_k)\right), \quad (13)$$

where $\lambda$ controls the penalty strength applied to clusters with low reasoning diversity.

### 3.3.4. SAMPLING STRATEGY

Let the total sampling budget be $M$. The target quota for cluster $k$ is defined as

$$q_k = \left\lfloor \frac{w_k}{\sum_{j=0}^{K-1} w_j} \cdot M \right\rfloor. \quad (14)$$

Any remaining quota caused by rounding is allocated sequentially to clusters with larger weights, ensuring

$$\sum_{k=0}^{K-1} q_k = M. \quad (15)$$

After determining the cluster-level quotas, we perform constrained sampling within each semantic cluster.

The goal of this stage is to meet the allocated quotas while suppressing template-like one-line proofs. We first identify one-shot proofs using regular expression rules and define the indicator variable

$$o_i = \mathbf{1}[P_i \text{ matches a one-shot proof pattern}]. \quad (16)$$

For each semantic cluster $\mathcal{C}_k$, the sampling procedure proceeds as follows:

- We randomly shuffle the samples in $\mathcal{C}_k$ to avoid ordering bias.

- Samples with $o_i = 0$ are selected sequentially until the cluster-level quota $q_k$ is reached or all non–one-shot samples in the cluster are exhausted.

- If the number of selected samples is still below $q_k$, one-shot samples are considered only when the following two constraints are simultaneously satisfied:

$$OS_k \leq \rho_{clu} \cdot q_k, \quad (17)$$

$$\frac{\sum_{x_i \in \mathcal{S}} o_i}{|\mathcal{S}|} \leq \rho_{glob}, \quad (18)$$

*Table 1.* **Comparison with prior neural theorem-proving methods on the miniF2F benchmark under the Isabelle environment.** We report proving success rates on miniF2F-test and miniF2F-valid. Previous approaches include DSP, Subgoal-XL, LEGO-Prover, Lyra, POETRY, and ProofAug. The best result is highlighted in bold.

| Method | Model | miniF2F-test | miniF2F-valid |
|---|---|---|---|
| DSP (Jiang et al., 2022) | CodeX | 39.3% | 42.6% |
| Subgoal-XL (Zhao et al., 2024) | Fine-tuned LLaMA-8B | 56.1% | 61.9% |
| LEGO-Prover (Wang et al., 2024b) | Mixed GPTs | 50.0% | 55.3% |
| Lyra (Zheng et al., 2024) | GPT-4 | 51.2% | 55.3% |
| POETRY (Wang et al., 2024a) | Fine-tuned ProofGPT (1.3B) | 42.2% | 42.2% |
| ProofAug (Liu et al., 2025) | deepseek-math-7b-base | 61.9% | 58.6% |
| ***Ours*** | deepseek-math-7b-base | **90.6%** | **86.5%** |

*Table 2.* **Comparison of different fine-tuning and inference strategies on the miniF2F benchmark.** The table reports Pass@k performance on miniF2F-test and miniF2F-valid under different training data scales and sampling budgets. We compare the untuned base model, fine-tuning on the raw corpus, fine-tuning on the high-quality dataset, and the complete framework with dynamic feedback-based prompt optimization.

| Method | Dataset Size | Sample Budget | miniF2F-test | miniF2F-valid |
|---|---|---|---|---|
| No Fine-Tuning | - | 32 attempts | 34.4% | 16.4% |
|  | - | 64 attempts | 42.6% | 37.7% |
| Raw Corpus SFT | $\approx 200,000$ | 32 attempts | 43.9% | 41.4% |
|  |  | 64 attempts | 51.2% | 49.2% |
| High-Quality Data SFT | $\approx 4,200$ | 32 attempts | 81.6% | 80.7% |
|  |  | 64 attempts | 84.8% | 82.4% |
| ***Full Framework*** | $\approx 4,200$ | 64 attempts | **90.6%** | **86.5%** |

where $\rho_{clu}$ and $\rho_{glob}$ denote the upper bounds on the proportion of one-shot samples at the cluster and global levels, respectively. Samples violating either constraint are skipped.

- Since these constraints may result in $|\mathcal{S}| < M$, we backfill from the remaining candidates, defined as

$$\mathcal{R} = \{x_i \in \mathcal{D}_{\text{mid}} \setminus \mathcal{S} \mid o_i = 0\}. \quad (19)$$

The candidates in $\mathcal{R}$ are sorted in descending order of their cluster entropy scores $H_{z_i}$, and backfilling prioritizes samples from high-entropy clusters:

$$\mathcal{S} \leftarrow \mathcal{S} \cup \text{Top}_{M-|\mathcal{S}|}(\mathcal{R}). \quad (20)$$

Here $\text{Top}_k(\mathcal{R})$ selects the $k$ samples in $\mathcal{R}$ with the highest cluster entropy scores $H_{z_i}$, where $H_{z_i}$ denotes the entropy of the semantic cluster to which $x_i$ belongs.

### 3.4. Dynamic Feedback-Based Prompt Optimization

Prior work has explored leveraging dynamic or verifiable signals for correction, including adaptive data augmentation (Vishwakarma & Mishra, 2023), neuro-symbolic self-correction with verifier feedback (Ambati, 2026), and verification-driven optimization with verifiable rewards (Liu et al., 2026); in contrast, we treat verifier feedback as a dynamic data signal and encode it directly at the prompt level for inference-time optimization.

At each round $r$, if verification fails, we extract the corresponding error message $E_r$ from the verifier. The subsequent generation then conditions on both the original problem description and the accumulated historical data signals, as illustrated in Figure 2, and is formulated as

$$\mathcal{P} = \mathcal{L} \oplus T_i \oplus \text{History}((P_k, E_k)_{k=1}^{r-1}). \quad (21)$$

Here, $\text{History}(\cdot)$ summarizes previous failed attempts by distilling key error signals and explicitly guiding the model to avoid repeatedly generating ineffective proof steps. In this way, historical error feedback is converted into informative constraints that progressively narrow the search space.

We further employ an early stopping mechanism: once a theorem is successfully proved, all remaining attempts are terminated to avoid redundant computation. In addition, we adopt a dynamic temperature scheduling scheme in which the sampling temperature increases linearly with the number of failed attempts:

$$T(r) = T_{\text{base}} + \alpha \cdot \frac{r}{N_{\text{max}}}. \quad (22)$$

This strategy encourages the model to favor high-confidence, conventional approaches in early rounds, while exploring more diverse reasoning strategies in later rounds if standard methods fail.

# 4. Experiments

## 4.1. Experimental Setup

### 4.1.1. BASELINES AND BENCHMARKS

**Benchmark** We evaluate our methods on the Isabelle portion of miniF2F (Zheng et al., 2022), a benchmark of Olympiad-level mathematics problems formalized across multiple proof assistants. Unless otherwise specified, we use miniF2F-test to refer to the Isabelle test set released in prior work (Jiang et al., 2021).

In our experiments, we extract only the formal lemma statements and provide them to the model via prompting, without using any informal proofs or proof sketches.

**Baselines** We compare our approach against prior work on the miniF2F-test, including DSP (Jiang et al., 2022), Subgoal-XL (Zhao et al., 2024), LEGO-Prover (Wang et al., 2024b), Lyra (Zheng et al., 2024), ProofAug (Liu et al., 2025), and POETRY (Wang et al., 2024a).

### 4.1.2. TRAINING AND EVALUATION CONFIGURATION

**Evaluation Setup** We evaluate all methods in Isabelle 2025 via the PISA environment (Jiang et al., 2021), with verification performed using 16 parallel PISA instances on a machine with 25 vCPUs. This setup enables efficient parallel proof checking while maintaining stable interaction with the theorem prover across multiple candidate generations.

**Fine-tuning and Inference** We perform supervised fine-tuning on DeepSeek-Math-7B-Base (Shao et al., 2024) using LoRA on the $4k$ high-quality training dataset, with a maximum sequence length of 4096. All experiments are conducted on two RTX 5090 GPUs (32GB). During inference, we adopt nucleus sampling with dynamic temperature scheduling implemented via vLLM (Kwon et al., 2023). To improve inference efficiency, candidate proofs within each round are generated and verified in parallel, while subsequent attempts are iteratively refined using verifier feedback from previous rounds.

All reported results are averaged over 10 independent runs to reduce randomness introduced by sampling-based decoding. Detailed hyperparameter settings are provided in Appendix A, while additional statistical results, including confidence intervals, are reported in Appendix C.

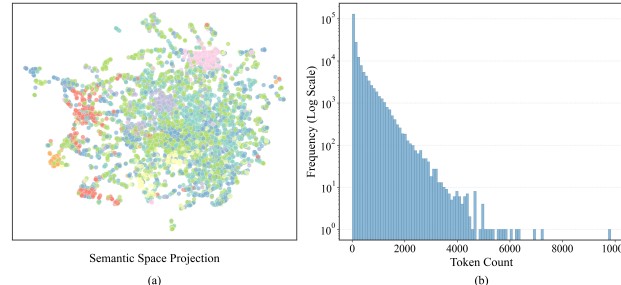

*Figure 3.* (a) Semantic cluster distribution in the raw corpus, showing severe imbalance. (b) Proof length distribution, exhibiting two extremes with very short and very long proofs.

## 4.2. Analysis of the Raw Corpus

We analyze the raw corpus (Hu et al., 2025), which consists of approximately 200,000 theorem–proof pairs collected from Isabelle/HOL and the Archive of Formal Proofs (AFP). This corpus provides broad coverage of formal mathematical reasoning patterns and serves as the primary source for our subsequent data selection experiments.

**Semantic Imbalance** As shown in Figure 3 (a), the semantic clusters in the raw corpus are imbalanced, with a small number of clusters accounting for the majority of samples.

**Length and Complexity Distribution** Figure 3 (b) shows that the corpus contains both extremely short and extremely long proofs, resulting in a highly uneven length distribution and reflecting substantial imbalance in proof complexity.

**Implications for Data-Efficient Training** Figure 4 shows that increasing the training data scale does not lead to monotonic improvements across the Pass@k metrics. In contrast, models trained on moderate-sized subsets achieve comparable and often more stable performance. Unless otherwise specified, all subsequent experiments use a training set of approximately $4k$ samples.

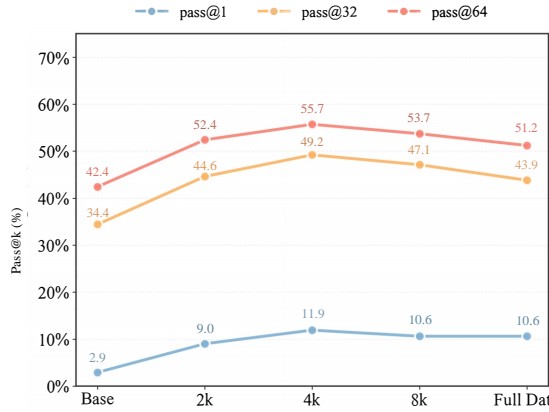

*Figure 4.* Pass@k performance on miniF2F-test for models fine-tuned on randomly constructed training subsets of different sizes.

## 4.3. Experimental Results

### 4.3.1. MAIN RESULTS

Table 1 compares our method with previous Isabelle-based neural theorem-proving approaches on the miniF2F benchmark. Our framework achieves the best overall performance, reaching 90.6% Pass@64 on miniF2F-test and substantially surpassing all previously reported results on miniF2F-Isabelle.

Table 2 further analyzes the effect of different training and inference strategies. Fine-tuning the base model on the curated high-quality dataset achieves 84.8% Pass@64, significantly outperforming fine-tuning on the full raw corpus despite using a much smaller training set, highlighting the importance of data quality over data quantity. We also report the corresponding performance on miniF2F-valid, where the high-quality dataset shows similarly strong generalization behavior. Building on this model, dynamic feedback-based prompt optimization further improves performance to 90.6% Pass@64, demonstrating the effectiveness of verifier-feedback-guided inference optimization.

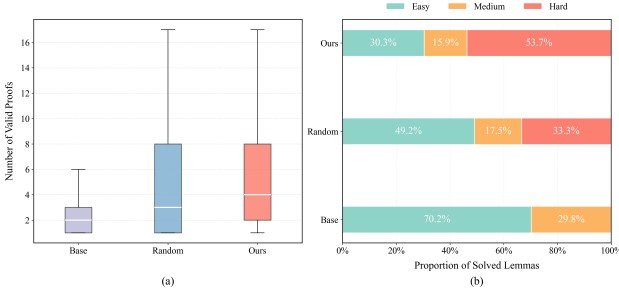

(a)                                (b)

*Figure 5.* **Fine-grained analysis of verification behavior on miniF2F-test.** (a) Number of verifier-accepted proofs per solved theorem under a fixed sampling budget. (b) Difficulty distribution of solved theorems across different models.

### 4.3.2. RESULTS ANALYSIS

Overall pass rates alone do not fully capture differences in model reasoning behavior. We therefore conduct a finer-grained analysis from two complementary perspectives.

**Proof diversity** As shown in Figure 5 (a), the base model typically produces only a small number of valid proofs per theorem. Random fine-tuning increases diversity for a limited subset of theorems, but the overall distribution remains highly skewed. In contrast, our approach consistently generates multiple structurally diverse valid proofs across a broader set of theorems, indicating stronger proof exploration capability rather than reliance on a single dominant proof pattern.

**Difficulty distribution** As shown in Figure 5 (b), the base model mainly succeeds on easy theorems, with limited cov-

erage of harder ones. To quantify theorem difficulty, we evaluate each theorem using three additional base models and categorize difficulty based on the frequency of successful proof generation. While random fine-tuning increases the number of solved medium and hard theorems, performance remains dominated by easy cases. In contrast, our approach substantially improves success on hard theorems, suggesting that the observed gains are not primarily driven by solving easy problems.

*Table 3.* Ablation Study of High-Quality Data Selection.

| Training Data | Qwen2.5 | DeepSeek |
|---|---|---|
| Base (No Fine-Tuning) | 20.9% | 34.4% |
| Random Selection | 39.3% | 49.2% |
| Complexity-Only | 44.3% | 63.9% |
| Complexity + Simple Clustering | 51.2% | 74.0% |
| *High-Quality Data Selection* | 58.2% | 81.6% |

## 4.4. Ablation Studies

We conduct ablation studies on the *High-Quality Data Selection Pipeline* and *Dynamic Feedback-Based Prompt Optimization* to assess the contribution of each component.

### 4.4.1. IMPACT OF THE DATA SELECTION PIPELINE

We evaluate four controlled settings that isolate different components of the data selection pipeline. Except for the base model, all models are fine-tuned on subsets constructed under different selection strategies. Specifically, we compare the following model variants: (1) **Base Model**, an unfine-tuned baseline; (2) **Random Selection**, fine-tuning on $M$ randomly sampled examples; (3) **Complexity-Only**, applying complexity-based filtering followed by random sampling; (4) **Complexity + Simple Clustering**, extending complexity-based filtering with K-Means clustering to improve semantic coverage.

As shown in Table 3, progressively incorporating these components leads to consistent Pass@32 performance gains, and their combination achieves the best results under this evaluation setting. This indicates that the effectiveness of the data selection pipeline stems from the cumulative contribution of complementary components rather than any single step.

We additionally report corresponding Pass@32 results on Qwen2.5-Coder-7B-Instruct (Hui et al., 2024). Although the overall performance improvements are less pronounced than those observed on DeepSeek-Math-7B, the same overall trend consistently holds across different data selection settings. This suggests that the effectiveness of the proposed high-quality data selection strategy generalizes across different backbone models.

*Table 4.* Ablation Study of Dynamic Feedback-Based Prompt Optimization.

| Method | Pass@64 |
|---|---|
| High-Quality Data SFT | 84.8% |
|     + *Fixed-Temperature Inference* | 89.8% |
|     + *Dynamic Feedback-Based Optimization* | 90.6% |

### 4.4.2. Impact of Dynamic Prompt Optimization

We ablate dynamic feedback-based prompt optimization, with results reported in Table 4. We compare inference variants that progressively incorporate fixed-temperature inference and dynamic feedback-based optimization.

Under a fixed inference budget of 64 attempts per theorem, fixed-temperature inference already yields substantial gains over single-shot generation by allowing limited exploration under a fixed sampling budget. Further introducing dynamic feedback-based prompt optimization leads to additional improvements, resulting in a total increase of 5.8% in Pass@64. This gain arises from explicitly conditioning each generation round on verifier feedback from previous failures, which guides the model to revise its reasoning strategy rather than repeatedly exploring semantically similar but invalid proof paths. As a result, the inference budget is used more effectively, with later attempts increasingly focused on unresolved and harder cases.

## 5. Discussion and Future Directions

Our work adopts a data-centric approach to neural theorem proving and demonstrates consistent performance improvements under the proposed data selection and prompt optimization strategy across multiple experimental settings and model configurations. We further observe that increasing the inference budget can continue to improve coverage, indicating that current limitations are largely associated with exploration dynamics rather than intrinsic model capability. Understanding this behavior and its implications for efficiency, convergence, and resource allocation is an important direction for future work.

In addition, it remains to be explored how the proposed approach generalizes to other interactive theorem provers such as Lean or Coq. Another important direction is to better understand the relationship between Isabelle's semantic structure and model behavior during proof generation. In particular, analyzing how semantic constraints, proof states, and verifier feedback contribute to the empirical improvements observed in our framework may provide insights into neural theorem proving. Finally, extending the current study beyond mathematical benchmarks toward project-level formal verification tasks is a natural next step.

## Acknowledgements

We would like to thank the anonymous reviewers for their valuable comments and suggestions. This work was supported by the National Natural Science Foundation of China under Grant No. 62132014 and the Zhejiang Provincial Natural Science Foundation of China under Grant No. LD24F020006.

## Impact Statement

This paper presents a data-centric approach for neural theorem proving in Isabelle, aiming to advance research in machine learning, reasoning, and formal verification. The proposed framework may benefit the development of automated proof generation and AI-assisted formal reasoning systems. We do not identify any immediate negative societal or ethical implications that require separate discussion.

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

# A. Training and Inference Configuration

## A.1. Experimental Parameter Settings

This section summarizes the hyperparameter settings used in our experiments for the $4k$ high-quality training dataset, including data selection, model training, and inference configurations. Unless otherwise specified, these parameters are used as the default settings in experiments based on the $4k$ dataset.

*Table A.* Hyperparameter settings for data selection, training, and inference.

| Parameter | Symbol | Value | Description |
|---|---|---|---|
| **High-Quality Data Selection & Dynamic Feedback–Based Prompt Optimization** | | | |
| Target subset size | – | 4271 | Size of the high-quality training subset |
| Retention ratio (mid-complexity) | $r$ | 0.5 | Retain samples in the middle complexity range |
| Proof complexity weight | $w_p$ | 0.6 | Weight of proof length |
| Theorem complexity weight | $w_t$ | 0.3 | Weight of theorem statement length |
| Strategy step weight | $w_s$ | 0.1 | Weight of tactic step count |
| Number of semantic clusters | $K$ | 50 | TF-IDF + KMeans clustering |
| TF-IDF maximum dimension | $d_{\max}$ | 60,000 | Upper bound of vocabulary size |
| TF-IDF minimum document frequency | minDF | 5 | Threshold for filtering low-frequency terms |
| Intra-cluster one-shot ratio | $\rho_{\mathrm{clu}}$ | 0.12 | Suppression of templated proofs within clusters |
| Global one-shot ratio | $\rho_{\mathrm{glob}}$ | 0.08 | Global one-shot constraint |
| Entropy penalty coefficient | $\lambda$ | 0.35 | Penalty for low-diversity clusters |
| **Training Hyperparameters** | | | |
| Maximum input length | – | 4096 | Maximum sequence length for training |
| Learning rate | – | $5 \times 10^{-5}$ | Peak learning rate |
| Number of training epochs | – | 5 | With early stopping |
| Warmup ratio | – | 0.03 | Linear warmup proportion |
| Effective batch size | – | 16 | Set via gradient accumulation |
| Optimizer | – | AdamW | Default AdamW configuration |
| Weight decay | – | 0.01 | AdamW default setting |
| Random seed | – | 42 | For reproducibility |
| **Inference and Sampling** | | | |
| Number of samples | $N$ | 64 | Maximum attempts per theorem |
| Samples per round | $B$ | 8 | Number of candidates generated per round |
| Base temperature | $T_{\mathrm{base}}$ | 0.6 | Initial sampling temperature |
| Temperature slope | $\alpha$ | 0.1 | Dynamic temperature adjustment |
| Top-$p$ | – | 0.95 | Nucleus sampling threshold |
| Top-$k$ | – | Not used | Full sampling space |
| Maximum generation length | – | 2048 | Maximum number of generated tokens during inference |

# B. Robustness of High-Quality Data Selection

This section analyzes the robustness of the proposed PSR-guided data selection pipeline with respect to key design choices. We focus on assessing whether the observed benefits of PSR are sensitive to reasonable variations in implementation details, including strategy extraction rules and clustering configurations. Overall, these analyses aim to verify that PSR exhibits stable data selection behavior rather than relying on brittle heuristics or finely tuned parameter settings.

## B.1. Robustness to Strategy Matching Order

To evaluate robustness to strategy extraction rules, we randomly shuffle the priority order of tactic labels used for strategy matching and re-run the PSR data selection pipeline. While the resulting cluster-level strategy entropy shows moderate variation (Pearson $r = 0.70$), the induced sampling quotas remain nearly identical (Pearson $r = 0.997$), and the final selected subsets exhibit a high overlap (Jaccard similarity $= 0.94$). These results indicate that PSR is largely insensitive to the specific ordering of tactic labels, and that its downstream data selection behavior remains stable under reasonable perturbations of the matching procedure.

## B.2. Sensitivity to Clustering Granularity

We further analyze the effect of clustering granularity by varying the number of semantic clusters used in PSR. Adjusting the number of clusters changes the semantic partitioning of the proof corpus, leading to different cluster-level statistics and selected subsets. Such variation is expected, as PSR explicitly enforces diversity through cluster-conditional measures rather than assuming invariance across granularities. In practice, the number of clusters controls the resolution at which semantic diversity is enforced, trading off semantic specificity against statistical stability. In this work, we select the clustering granularity based on this trade-off.

## C. Additional Experimental Results

*Table C.* Additional Statistical Results.

| Method | Pass@32(mean±std) | 95% CI | Pass@64(mean±std) | 95% CI |
| --- | --- | --- | --- | --- |
| Base (No Fine-Tuning) | 34.4±1.3 | [33.5,35.4] | 42.6±1.3 | [41.7,43.5] |
| Raw Corpus SFT | 43.9±1.0 | [43.1,44.6] | 51.2±1.4 | [50.3,52.2] |
| High-Quality Data SFT | 81.6±1.0 | [80.7,82.2] | 84.8±1.0 | [84.1,85.5] |
| Full Framework | - | - | 90.6±3.3 | [88.2,93.0] |

Table C reports the mean performance and corresponding 95% confidence intervals over 10 independent runs. The relatively small confidence intervals indicate that the proposed framework exhibits stable performance across different sampling runs. In particular, the improvements over the raw corpus baseline remain consistent under repeated evaluation.

## D. Mechanism of Dynamic Feedback-Based Prompt Optimization

This section provides a more detailed description of the inference-time mechanism of Dynamic Feedback-Based Prompt Optimization. Unlike independent repeated sampling, which treats each proof attempt as an isolated generation, our method maintains a history-aware prompt across inference rounds and uses verifier feedback as an explicit dynamic signal to guide subsequent generations.

For each theorem, the model is given a maximum budget of 64 proof attempts. These attempts are organized into multiple rounds, with 8 candidate proofs generated in parallel at each round. After generation, all candidates in the current round are verified by Isabelle through the PISA interface. If any candidate is successfully verified, the theorem is marked as solved, and no further attempts are generated for that theorem. This early-stopping strategy avoids unnecessary verification and reduces the total number of generation attempts.

If all candidates in a round fail verification, the corresponding Isabelle error messages are collected and summarized into a compact feedback signal. The feedback typically contains information such as unresolved subgoals, failed tactic applications, type mismatches, or invalid proof steps. Rather than directly appending all raw error messages to the prompt, we extract the most informative diagnostic signals and combine them with a summary of previous failed attempts. This forms a structured feedback history that is incorporated into the prompt for the next round.

Formally, at round $r$, the prompt is constructed from three components: the original theorem statement, the proof-generation instruction, and the accumulated feedback history from previous rounds. This history encourages the model to avoid repeating previously invalid proof patterns and to revise its reasoning strategy in response to verifier-provided constraints. In this way, verifier feedback is treated as a dynamic data signal during inference, progressively narrowing the search space and guiding the model toward more promising proof directions.

In addition to feedback conditioning, we apply dynamic temperature scheduling across rounds. Early rounds use a lower temperature to encourage conventional and high-confidence proof strategies, while later rounds gradually increase the temperature after repeated failures. This design balances exploitation and exploration: the model first attempts standard proof patterns, and then explores more diverse strategies when earlier attempts fail.

Overall, Dynamic Feedback-Based Prompt Optimization differs from standard Pass@$k$ sampling in two aspects. First, the attempts are not independent, since later generations are conditioned on verifier feedback from previous failures. Second, the inference process is adaptive, since solved theorems are removed from further generation through early stopping, and unsolved theorems receive increasingly targeted feedback. This mechanism allows the fixed inference budget to be used more efficiently, reducing redundant exploration of similar invalid proof paths.

## E. Data Contamination Analysis

To ensure that the reported performance is not affected by train-test contamination, we conduct an additional contamination analysis between the training corpus and the miniF2F-test benchmark. Since the miniF2F-test does not provide reference proofs as model input, we focus on the theorem-statement-level overlap analysis.

The training corpus consists of theorem-proof pairs collected from Isabelle/HOL and the Archive of Formal Proofs (AFP), while evaluation is performed on the miniF2F-test.

We perform contamination detection using multiple matching strategies. First, we conduct exact string matching on theorem statements after removing duplicated whitespace and normalizing formatting symbols. Second, we apply canonical normalization, including lowercasing, whitespace normalization, and removal of non-alphanumeric symbols, followed by exact matching on the normalized statements.

To further identify potential near-duplicate cases, we additionally compute sequence similarity scores over normalized theorem statements. Exact matching is performed after whitespace normalization. To account for differences in declaration names and theorem keywords, we also remove Isabelle declaration headers, such as `lemma name:` and `theorem name:`, and compare the remaining theorem bodies. Potential near-duplicate matches are further identified using character-level sequence similarity with a threshold of 0.8.

Across all matching strategies, we do not observe any exact or near-duplicate overlap between the two datasets. These results suggest that the reported improvements are unlikely to arise from memorization of overlapping theorem statements.

