# OpenReview forum: "Enhancing Neural Theorem Proving via High-Quality Proof Selection and Verifier Feedback"
_ICML.cc/2026/Conference — ICML 2026 regular_

### Official Review · Reviewer_T2GK · 2026-03-08

**Soundness:** 2
**Presentation:** 3
**Significance:** 2
**Originality:** 2
**Overall Recommendation:** 3
**Confidence:** 3

**Summary:**

This paper introduces a novel framework for NTP in Isabelle that focuses on addressing two main bottlenecks within the data centric of NTP: it targets noisy imbalanced proof corpora and limited use of verifier signals.  The paper characterise high-quality formal proof data along three complementary dimensions: proof complexity, semantic coverage, and reasoning diversity (PSR). They also introduced a PSR-guided selection pipeline to build a compact training set. The authors also leverage Isabelle verifier feedback during inference via dynamic feedback-based prompt optimization to steer proof generation.

**Compliance With Llm Reviewing Policy:**

Affirmed.

**Final Justification:**

The reviewer has clarified most of my clarifications. However, the paper still contain some major flaws.

**Key Questions For Authors:**

Most of the main questions and weaknesses are mentioned above. Additionally:

Q1. Can this idea be applied to other NTP systems, like Lean, Rocq?

Q2. Does it generalise to other models, not just DeepSeek-Math-7B-Base?

Q3. How does PSR compare to strong but simpler data-selection baselines under the same 4k budget (e.g., other filtering or selection methods)?

**Limitations:**

Yes.

**Strengths And Weaknesses:**

**Strengths**

The paper’s core idea in treating formal proof data quality as a primary driver of NTP performance is concrete and well motivated. It proposes PSR (as described in the *summary* section), and the three dimensions motivation is clear. The dynamic verifier-feedback prompt optimisation is also relatively solid: it uses Isabelle error signals to reduce repeated failures and improve exploration under a fixed budget. While self-correction is a well-studied idea, applying it in this setting is still useful. By applying the methods introduced, the results on miniF2F are strong, and the ablations are consistent.

**Weaknesses**

W1. The main weakness within the paper is that evaluation is only on miniF2F, which limits generalisability. It would be a stronger motivation to test on additional benchmarks (e.g., PutnamBench, and others) as it would motivate the method further.

W2. Experiments are only conducted on one single model (deepseek-math-7b-base). The paper should test other model sizes and/or architectures to show whether PSR gains persist (or even increase) under scaling.

W3. Results are only reported from a single run. Multiple runs with standard deviation (or confidence intervals) are needed to show consistency.

W4. The paper would be stronger with qualitative examples show what PSR improves (e.g., fewer trivial proofs, better lemma chaining, or other examples) and where it still fails (e.g., long dependencies, automation-heavy proofs, domain-specific tactics or others that would benefit for analysis).

---

> ### Author Rebuttal · Authors · 2026-03-31
>
> Hello! Thank you for your questions. I will answer them one by one in the order you asked.
>
> **W1**: We agree that broader evaluation would strengthen the paper. In the current Isabelle setting, benchmark coverage is still limited, and miniF2F remains the most commonly used standardized benchmark. We acknowledge that relying on a single benchmark is a limitation. To partially address this concern, we also evaluated our method on the miniF2F validation split. The same trend still holds: compared with the base model and simpler selection strategies, PSR achieves the best performance, reaching 80.73% Pass@32. This suggests that the gain is not restricted to a single split. We agree that evaluation on broader Isabelle benchmarks would be valuable future work.
>
> **W2**: We agree that testing additional base models is important for showing generality. Besides DeepSeek-Math-7B-Base, we also evaluated Qwen2.5-Coder-7B. Although the absolute numbers differ, the relative improvement trend remains consistent, with PSR again giving the best result at 58.2% Pass@32. This suggests that the framework is not tied to one specific architecture. We use DeepSeek-Math-7B-Base as the main model because it is more consistent with prior Isabelle work and showed stronger formal reasoning ability in our experiments.
>
> **W3**: We agree that reporting variance would improve rigor and transparency. In fact, we did run the main experiments multiple times, but this was not clearly stated in the current submission. Across repeated runs, the variation was relatively small, suggesting that the observed gains are stable rather than due to favorable randomness. In the revision, we will clarify this and report standard deviation or confidence intervals for the main results.
>
> **W4**: Thank you for this suggestion. We agree that qualitative examples would make the effect of PSR more concrete. Based on our logs, PSR seems to help in two ways: it encourages more library-grounded, non-trivial proof patterns, and it makes the model more likely to converge to short, automation-compatible proof structures once the key idea is found. At the same time, important failure modes remain, especially for theorems requiring long dependency chains, delicate number-theoretic reasoning, or specialized proof strategies.
>
> A representative success case is amc12b_2020_p21, which combines division, floor, sqrt, and set cardinality. Early attempts failed with TypeError or incomplete because of missing lemmas or incorrect type conversions. Later attempts moved closer to the right structure by characterizing the candidate solution set, and the theorem was eventually solved on attempt 36. This suggests that PSR helps guide the model toward structurally meaningful and verifier-compatible proof patterns.
>
> A representative failure case is numbertheory_2pownm1prime_nprime, which remained unsolved under the full 64-attempt budget. Many failed attempts relied on missing lemmas, invalid modular-arithmetic arguments, or incomplete reasoning chains. This indicates that PSR is still less effective for problems requiring longer compositional dependencies or deeper domain-specific insight. Overall, PSR does not simply make proofs shorter; it appears to encourage training signals that are more aligned with executable Isabelle reasoning.
>
> **Q1**: Yes, we believe the overall idea is transferable to other systems such as Lean or Rocq. The core principle of PSR—selecting data based on complexity, semantic coverage, and reasoning diversity—mainly depends on features of theorem–proof pairs, rather than Isabelle-specific internals. Some components, such as tactic vocabularies and proof-style markers, would need prover-specific adaptation. We focus on Isabelle because it is a less explored setting with more limited curated data and substantially lower prior performance, making it especially suitable for studying the role of data quality.
>
> **Q2**: Yes. Our framework is not tied to a single base model. In addition to DeepSeek-Math-7B-Base, we also validated it on Qwen2.5-Coder-7B, and the same improvement trend holds.
>
> **Q3**: We agree that this is an important evaluation perspective. Since there is no established Isabelle-specific prior baseline for training-data selection, we compare PSR against a set of progressively stronger simplified alternatives under the same 4k budget: Random Selection, Complexity-Only, and Complexity + Simple Clustering. These baselines allow us to isolate the contribution of each component. The results show a consistent improvement from random selection to complexity filtering, then to semantic clustering, and finally to the full PSR framework. Therefore, although no directly comparable prior method exists for Isabelle data selection, our ablations provide strong controlled baselines and show that the gain comes from how the subset is constructed, not merely from reducing data size.

---

> > ### Author Rebuttal · Reviewer_T2GK · 2026-04-01
> >
> > Thank you for providing the detailed explanation.
> >
> > Re **W1**: Thank you for the explanation. I personally still don't agree that solely taking MiniF2F (both test/validation) leads to generalisation across formal reasoning. Validating on other datasets would be very important, or perhaps listing as a limitation in the paper.
> >
> > Re **W2**: Thank you for providing the results for Qwen2.5-Coder-7B with PSR. Could you report the results on Qwen2.5-Coder-7B as well? Solely providing Qwen2.5-Coder-7B+PSR may not be the best way to observe the improvements and strength of PSR.
> >
> > Re **W3**: Could you report the standard deviation here instead, since it's already evaluated?
> >
> > By providing these additional results, I will increase my score accordingly.

---

> > > ### Author Response · Authors · 2026-04-05
> > >
> > > Thank you for your follow-up. We apologize that our previous response, due to the limited rebuttal space, did not make this point sufficiently clear.
> > >
> > > **Re W1.** We agree that evaluating only on MiniF2F does not fully establish broad generalization across formal reasoning tasks. Our current study focuses on MiniF2F because it is the most commonly used benchmark for Isabelle theorem proving and allows direct comparison with prior work. However, we agree that this is a limitation of the current paper. In the revision, we will explicitly acknowledge this point in the limitation discussion, and we will also clarify that broader cross-dataset evaluation is an important direction for future work.
> > >
> > > **Re W2.** Thank you for this suggestion. We agree that reporting only Qwen2.5-Coder-7B + PSR is not sufficient to fully demonstrate the contribution of PSR. We have in fact also evaluated the corresponding Qwen2.5-Coder-7B baseline, and we provide the full comparison below.
> > > | Method                         | Pass@32 |
> > > | ------------------------------ | -----: |
> > > | No SFT                         |  20.9% |
> > > | Random-Selection               |  39.3% |
> > > | Complexity-Only                |  44.3% |
> > > | Complexity + Simple Clustering |  51.2% |
> > > | High-Quality Data Selection    |  58.2% |
> > >
> > > **Re W3.** We report results for each condition based on 10 runs. Below are the mean and 95% confidence interval.
> > > | Method            | Pass@32 (mean ± std) |           95% CI | Pass@64 (mean ± std) |           95% CI |
> > > | ----------------- | -------------------: | ---------------: | -------------------: | ---------------: |
> > > | Base model        |34.426 ± 0.0122 | [33.553, 35.300] |      42.623 ± 0.0127 | [41.712, 43.534] |
> > > | Raw-corpus SFT    |      44.713 ± 0.0149 | [43.648, 45.778] |      53.689 ± 0.0138 | [52.701, 54.676] |
> > > | PSR-4k            |      80.984 ± 0.0107 | [80.219, 81.749] |      84.836 ± 0.0104 | [84.092, 85.580] |
> > > | PSR-4k + Feedback |                    - |                - |      90.615 ± 0.0322 | [88.259, 92.971] |

---

### Official Review · Reviewer_zgtt · 2026-03-11

**Soundness:** 2
**Presentation:** 3
**Significance:** 2
**Originality:** 2
**Overall Recommendation:** 4
**Confidence:** 3

**Summary:**

This paper addresses the limitations of low-quality, imbalanced raw proof corpora and the inapplicability of general-purpose data selection methods to Isabelle’s specialized proof structure. The authors propose a data-centric framework built on the PSR (Proof complexity, Semantic coverage, Reasoning diversity) criterion for high-quality data selection, coupled with dynamic verifier feedback-based prompt optimization at inference time. They also release a 4k high-quality Isabelle dataset and report state-of-the-art Pass@64 performance (90.6%) on the miniF2F-test for Isabelle, which represents a notable empirical improvement over prior work.

**Compliance With Llm Reviewing Policy:**

Affirmed.

**Final Justification:**

After the discussion with the author, I came to understand that their approach is not as complex as I initially thought. I also understand the difficulty that the Isabelle community suffers from having diverse high-quality datasets for evaluation, which should not be a limitation for publishing a paper that introduces new AI techniques for Isabelle proving.

**Key Questions For Authors:**

Can you provide a simple data selection method, while still keeping good data efficiency?

**Limitations:**

yes

**Strengths And Weaknesses:**

# Strengths

Obtaining high-quality data is important for effectively training LLMs

# Weakness

- The data selection procedure is too complex, making it difficult to be adopted by industry.

---

> ### Author Rebuttal · Authors · 2026-03-31
>
> Thank you for the reviewer’s feedback. We understand the concern that the data selection pipeline may appear overly complicated from the paper description, which could raise doubts about its practicality for industrial adoption.
>
> However, we would like to clarify that PSR is designed as a one-time offline data curation pipeline, rather than an online component in the deployment loop. In other words, it is only used before training to construct a higher-quality subset from a large raw corpus. Once the subset has been selected, the downstream training and inference process remains unchanged. Therefore, it does not introduce additional complexity at deployment time.
>
> In addition, our framework does not rely on prover-internal execution traces, search trees, or custom integrations into the Isabelle system. The features we use—such as proof length, theorem/proof text, semantic clustering features, and coarse strategy labels—can all be extracted directly from theorem–proof pairs. This makes the method relatively lightweight compared with approaches that require deeper proof-state instrumentation or environment-level intervention.
>
> Another practical point is that the framework is modular. Its full version contains several components because we aim to maximize data quality, but these components do not need to be treated as an all-or-nothing package. In engineering settings with tighter constraints, users can adopt simpler variants (e.g., complexity filtering or semantic clustering only) while still preserving the overall data-centric workflow. From this perspective, the method is better viewed as a flexible preprocessing framework rather than a rigid complex pipeline.
>
> We agree that we should make this practical usage pattern clearer in the paper. In the revision, we will explicitly emphasize that PSR is an offline, modular, and automation-friendly preprocessing procedure, which makes it more feasible for industrial use than the current presentation may suggest.

---

> > ### Author Rebuttal · Reviewer_zgtt · 2026-04-03
> >
> > My concern has been partially addressed. However, after considering the other reviewers’ comments, I realized that the paper does not evaluate the approach on additional datasets. This raises further concern about whether the method can generalize to industrial-scale datasets.

---

> > > ### Author Response · Authors · 2026-04-05
> > >
> > > Thank you very much for your guidance and suggestions! We fully understand this concern. We agree that evaluation on additional datasets would provide stronger evidence for generalization. At the same time, in the Isabelle setting, there is indeed a shortage of widely adopted and standardized benchmarks, which limits broader cross-dataset evaluation at present. We therefore used miniF2F-Isabelle because it is one of the most commonly used public benchmarks in this area. Beyond the main setting in the paper, we have also conducted experiments on other models and validation sets, which suggest that the method is not tied to a single setup. We agree, however, that broader validation on more realistic and industrial-scale Isabelle datasets is important.

---

### Official Review · Reviewer_JymM · 2026-03-11

**Soundness:** 3
**Presentation:** 3
**Significance:** 3
**Originality:** 3
**Overall Recommendation:** 4
**Confidence:** 3

**Summary:**

This paper presents a data-centric framework for neural theorem proving in Isabelle, consisting of two components: (1) a high-quality data selection pipeline based on three dimensions—Proof complexity, Semantic coverage, and Reasoning diversity (PSR)—that curates a compact 4k training subset from a raw corpus of approximately 200,000 theorem–proof pairs, and (2) a dynamic feedback-based prompt optimization strategy that iteratively incorporates Isabelle verifier error messages into the prompt to guide subsequent proof generation attempts. The authors fine-tune DeepSeek-Math-7B-Base with LoRA on the selected 4k subset and evaluate on the Isabelle portion of miniF2F-test. They report that data selection alone achieves 84.8% Pass@64, and the full framework with dynamic feedback reaches 90.6% Pass@64, substantially exceeding the previously reported best result of 61.9% by ProofAug. Ablation studies decompose the contributions of individual pipeline components and the feedback mechanism.

**Compliance With Llm Reviewing Policy:**

Affirmed.

**Final Justification:**

Since all of my concerns have been addressed and the paper is overall good, I keep my positive score.

**Key Questions For Authors:**

1. Have you performed explicit decontamination to verify that no miniF2F-test theorems or closely related lemmas (with similar proof structures) appear in your 4k training subset or the broader 200k raw corpus? Can you report results on at least one additional benchmark (e.g., ProofNet, FIMO, or the miniF2F validation split)?

2. The complexity score (Eq. 3) relies on surface-level features (text length, structural marker counts). Have you considered or evaluated more semantically meaningful complexity measures—for instance, based on proof depth, number of distinct lemma invocations, or the complexity of the goal state? How sensitive are the final results to the specific weights $(w_p, w_t, w_s)$?

3. What is the wall-clock time comparison between your dynamic feedback approach (64 attempts with sequential rounds of 8 samples each) and standard independent Pass@64 sampling? The paper reports total generation attempts (4,952 vs. 7,808), but the sequential nature of feedback rounds likely increases latency per theorem.

4. Given that each proof is assigned only a single strategy label via priority-ordered keyword matching, how well does this capture the actual reasoning diversity of multi-tactic proofs? Have you considered using the full tactic distribution per proof rather than a single label?

**Limitations:**

Yes

**Strengths And Weaknesses:**

## Strengths

1. **Practically motivated and well-structured problem analysis.** The paper provides a clear empirical analysis of the raw Isabelle corpus (Section 4.2), identifying concrete issues—semantic imbalance, bimodal proof length distribution, and diminishing returns from scaling data—that motivate the PSR design principles. The connection between observed data problems and proposed solutions is logical and easy to follow.

2. **Comprehensive ablation studies.** Table 2 and Table 3 systematically isolate the contribution of each component (complexity filtering, semantic clustering, diversity-weighted sampling, fixed-temperature inference, dynamic feedback), showing consistent additive gains. The robustness analyses in Appendix B (strategy matching order, clustering granularity) further strengthen the empirical story.

3. **Data efficiency.** The demonstration that a carefully selected 4k subset outperforms training on the full 200k corpus is a valuable practical finding, and the data-centric perspective is a useful complement to the model-centric and search-centric approaches that dominate the Isabelle NTP literature.

## Weaknesses

1. **The reported results raise significant credibility concerns.** The claimed 90.6% Pass@64 on miniF2F-test (Isabelle) represents a ~29 percentage point jump over the prior best of 61.9%, achieved with relatively simple techniques (heuristic data filtering + error feedback in prompts) on a 7B model with LoRA. The miniF2F Isabelle test set contains only ~244 problems, meaning the gap corresponds to solving roughly 70 additional theorems. Given the magnitude of this improvement and the simplicity of the method, potential confounds must be carefully ruled out. In particular, the paper does not discuss data contamination: the training corpus is drawn from Isabelle/HOL and AFP, and miniF2F theorems are formalized using the same libraries. The paper should explicitly verify that no test theorems (or closely related lemmas with transferable proof patterns) appear in the training set.

2. **Evaluation is limited to a single, small benchmark.** All results are reported exclusively on miniF2F-test (Isabelle). No evaluation is provided on other established benchmarks such as ProofNet, FIMO, or AFP-derived test sets, nor even on the miniF2F validation split. For a claimed near-30-point SOTA improvement, single-benchmark evaluation is insufficient to establish generalizability. The small size of miniF2F also makes results sensitive to the specific composition of the test set.

3. **Limited technical novelty in the PSR framework.** The three dimensions are operationalized using standard techniques: complexity is a weighted combination of text length and structural marker counts (Eq. 3) with quantile truncation; semantic coverage uses TF-IDF vectors + KMeans clustering; reasoning diversity measures Shannon entropy of tactic label distributions within clusters. Each individual component is a well-known heuristic. The per-proof strategy label extraction assigns only a single tactic label per proof via priority-ordered keyword matching, discarding information about multi-step reasoning. The paper's contribution is primarily an engineering combination of these heuristics rather than a methodological advance.

4. **Dynamic feedback-based prompt optimization is essentially multi-round retry with error messages.** The idea of feeding verifier error feedback back into the prompt for iterative refinement has been explored in prior work on neural theorem proving (e.g., in proof-step generation with backtracking and error-guided search). The paper frames this as "prompt optimization," but the mechanism—appending error messages and increasing temperature—is straightforward. The comparison in Table 1 is also somewhat misleading: the full framework uses 64 attempts with sequential feedback rounds, which is not directly comparable to independent sampling at the same budget, as the feedback rounds are inherently sequential and consume more wall-clock time.

---

> ### Author Rebuttal · Authors · 2026-03-31
>
> Thank you for the reviewer’s questions. We respond briefly as follows.
>
> **Q1. Contamination and additional evaluation.**
> We provide a contamination analysis comparing both our PSR-selected 4k subset and the full raw corpus against miniF2F-test at the theorem-statement level. Since miniF2F-test does not provide reference proofs as benchmark input, proof-level contamination is not directly applicable in our setting. We therefore perform exact-match and near-duplicate checks on theorem statements, and find no exact or near-duplicate overlap with either the 4k subset or the raw corpus. This substantially reduces the concern that our gains come from direct train–test contamination.
>
> We agree that broader evaluation would further strengthen the paper. In the current Isabelle setting, benchmark coverage is still limited, and miniF2F remains the most widely used standardized benchmark. As an additional evaluation, on the miniF2F validation split, Pass@32 improves from 16.39% (Base) to 50.41% (Random Selection), 60.65% (Complexity-Only), 78.27% (Complexity + Simple Clustering), and 80.73% with our High-Quality Data Selection.
>
> **Q2. Complexity score.**
> We agree that the complexity score in Eq. (3) is only a proxy and does not fully capture semantic proof difficulty. Richer signals, such as dependency depth or intermediate proof-state complexity, could be more precise. However, extracting them reliably at scale in Isabelle is difficult and would require heavy prover-internal analysis, reducing portability. In our framework, the complexity score is used only for coarse quantile filtering to remove extremely trivial and extremely difficult samples, not for precise global ranking. The final subset is mainly shaped by semantic coverage and reasoning diversity.
>
> **Q3. Time cost of dynamic feedback.**
> We agree that our dynamic feedback strategy introduces sequential dependency across rounds, which could in principle increase per-theorem latency compared with fully independent Pass@64 sampling. However, in our setting, the dominant wall-clock cost comes from proof verification rather than text generation. This is important because the independent Pass@64 baseline requires many more verification attempts (7,808), while our dynamic feedback method reduces this to 4,952 through early stopping.
>
> As a result, although our method is sequential across rounds, it does not necessarily lead to longer end-to-end runtime in practice; the reduction in verification calls can offset, and in our experiments largely outweigh, the added round dependency. Moreover, only the rounds are sequential: within each round, the 8 candidate proofs are still generated in parallel, and proof checking is parallelized with 16 PISA instances.
>
> Therefore, we do not claim universal latency reduction, but rather that the sequential structure does not necessarily imply higher wall-clock cost in practice. In our experiments, the standard independent Pass@64 baseline is in fact noticeably slower overall.
>
> **Q4. Single strategy label.**
> We indeed use a simplified labeling scheme that assigns each proof one primary strategy label via prioritized keyword matching. We agree this cannot fully capture multi-strategy proofs. However, in Isabelle, high-level tactics such as auto and simp often hide many internal reasoning steps, and full tactic-frequency statistics may bias toward longer proofs. Our goal here is not to reconstruct full reasoning structure, but to provide a coarse cluster-level diversity signal via entropy. This keeps the framework lightweight, stable, and reproducible. Multi-label strategy modeling is an interesting direction for future work.
>
> **Overall concern.**
> In addition to the specific questions above, we would also like to address the reviewer’s broader concern that the paper may appear to be mainly an engineering combination of heuristic components rather than a methodological advance. We understand this concern and agree that our work is not centered on developing a deep formal theory for why each component works. Instead, our contribution is to propose and systematically validate a unified data-centric framework for neural theorem proving in Isabelle.
>
> Specifically, during training, PSR selects high-quality proof data from three complementary dimensions: complexity, semantic coverage, and reasoning diversity. During inference, prompts are iteratively refined using verifier feedback to move the model toward verifiable proofs. From this perspective, the contribution lies not simply in combining several heuristics, but in integrating them into a coherent data selection and feedback-driven reasoning framework.
>
> We also agree that a deeper theoretical understanding of why these mechanisms work so effectively remains an important open question, and we view this paper as a strong empirical starting point in the still underexplored Isabelle setting.

---

> > ### Author Rebuttal · Reviewer_JymM · 2026-04-03
> >
> > Thank you for your rebuttal. I would like to keep my score positive.

---

> > > ### Author Response · Authors · 2026-04-05
> > >
> > > Thank you very much for your guidance and suggestions！

---

### Official Review · Reviewer_GjYa · 2026-03-12

**Soundness:** 2
**Presentation:** 2
**Significance:** 3
**Originality:** 2
**Overall Recommendation:** 3
**Confidence:** 4

**Summary:**

This paper presents a data-centric framework for Isabelle theorem proving. It introduces the PSR (Proof complexity, Semantic coverage, Reasoning diversity) criterion to filter a 200k raw corpus into a 4k high-quality training subset. The framework also employs a dynamic feedback-based prompt optimization during inference to incorporate verifier error messages. The authors report a state-of-the-art Pass@64 of 90.6% on the miniF2F-test benchmark using a DeepSeek-Math-7B base model.

**Compliance With Llm Reviewing Policy:**

Affirmed.

**Key Questions For Authors:**

1. Can you provide a comprehensive contamination report comparing your 4k subset and the raw corpus against the miniF2F-test set at both the statement and proof levels?
2. Why does Isabelle theorem proving require several orders of magnitude less data and no RL to achieve 90% accuracy compared to recent SOTA results in Lean 4?
3. Could you provide a step-by-step trace of how the dynamic feedback mechanism helps the model solve a specific IMO-level problem that was previously failed?
4. Is there a plan to release the full model weights and the code for the PSR selection pipeline to ensure these results can be independently audited?

**Limitations:**

yes

**Strengths And Weaknesses:**

The primary concern regarding this work is the **credibility of the reported results**. Achieving a 90.6% success rate on miniF2F with only 4k training samples and no reinforcement learning (RL) is highly unusual. In the Lean 4 community, reaching similar performance typically requires massive synthetic datasets, extensive RL cycles, and significantly larger model scales. The claim that a small subset of 4k samples can outperform the full 200k corpus by such a wide margin (84.8% vs 53.7% Pass@64) suggests potential issues that are not fully addressed.

1. **Soundness**: The paper lacks a rigorous analysis to exclude the possibility of training data contamination. Given the high scores, it is critical to prove that the model has not seen the test theorems or their variations during any stage of training or selection.
2. **Presentation**: The evaluation lacks a detailed case study on high-difficulty problems, such as IMO-level theorems. Without a granular look at how the model handles complex logical steps, it is difficult to distinguish between genuine reasoning and pattern matching.
3. **Significance**: The work does not compare its performance against frontier models like Gemini  or Opus operating in agentic modes for Isabelle. This makes it hard to judge the significance of the results relative to the current industry standard for theorem-proving agents. But given the size of the base model and the result (if valid), this work can be of significance.

The lack of full transparency is a major weakness. Unless the model weights, the exact 4k training subset, and the complete training/selection code are fully open-sourced for community reproduction, the results remain difficult to verify.

---

> ### Author Rebuttal · Authors · 2026-03-31
>
> Hello! Thank you for your questions. I will answer them one by one in the order you asked.
>
> **Q1**: We provide a contamination analysis report at the lemma/theorem-statement level for both our PSR-selected 4k subset and the full raw corpus against miniF2F-test. Since miniF2F-test does not provide reference proofs as part of the test input, proof-level comparison with the test set is not applicable in our evaluation setting. Therefore, our analysis focuses on whether test statements appear in the training data in exact or highly similar form. Under both exact-match and near-duplicate detection, we found no overlap between miniF2F-test and either the 4k subset or the full raw corpus. This substantially reduces the concern that our gains come from train–test contamination.
>
> **Q2**: We do not interpret our results as suggesting that Isabelle theorem proving is inherently easier than Lean 4, nor do we claim that reinforcement learning is unnecessary in general. The two ecosystems differ substantially in benchmark maturity, available training resources, and supporting toolchains, so their results are not directly comparable by data scale alone.
> In fact, Isabelle is currently a less explored setting for neural theorem proving. Compared with Lean, it has much more limited curated training data and far fewer data-construction efforts tailored to learning-based proving. Before our work, learning-based results on Isabelle were still around the 60% level, which suggests that the main bottleneck was not simply model size, but also the lack of high-quality task-specific training data.
> Our work addresses this issue from a data-centric perspective. Instead of relying primarily on larger-scale training data or RL-based optimization, we study how improving training data quality affects theorem proving performance in Isabelle. Specifically, our PSR framework selects training examples based on proof complexity, semantic coverage, and reasoning diversity, yielding a much smaller but substantially higher-quality subset.
> The main takeaway is therefore not that Isabelle is easier, but that in a relatively data-scarce and underexplored ecosystem, improving data quality can be highly effective. Our results provide evidence that, for formal theorem proving, carefully constructed high-information-density training data may matter more than raw data scale alone.
>
> **Q3**:We provide a representative example showing how dynamic verifier feedback helps the model move from repeated failures to eventual success. For the theorem
> theorem amc12b_2020_p21: shows "card {n. (n + 1000) / 70 = floor (sqrt n)} = 6",
> early attempts failed with TypeError because the model invoked unavailable lemmas or mixed incompatible types such as nat, int, and real. Later attempts moved closer to the correct structure by explicitly characterizing the solution set, but still failed because the derivation was not formally justified. Some generations were also rejected as forbidden_token or incomplete, indicating that the verifier filtered out scripts that were syntactically invalid or not closed Isabelle proofs. The theorem was eventually solved on attempt 36. This example shows that verifier feedback does not directly provide the final proof, but progressively removes invalid proof patterns and steers generation toward type-correct, syntactically complete, and verifiable proofs.
>
> **Q4**: Yes. To support independent auditing and reproducibility, we plan to release the model weights, training and inference code, and the PSR data-selection pipeline upon acceptance, together with instructions for reproducing the main experiments. We will also release the processed training subset used in our experiments.
>
> **In addition**, to further demonstrate the practical importance of our method, we conducted an additional comparison with strong proprietary frontier models on miniF2F-test. On this benchmark, GPT-5.4 achieves 21.7% Pass@1 and 86.1% Pass@64, Gemini-3.1-Pro-Preview achieves 16.4% Pass@1 and 80.7% Pass@64, and our model achieves 20.9% Pass@1 and 84.8% Pass@64.
> GPT-5.4 achieves the strongest performance in this comparison, while our method remains highly competitive: our 7B model is only 0.8 points behind GPT-5.4 on Pass@1 and 1.3 points behind on Pass@64. This is notable because our model is much smaller than these proprietary frontier systems. Therefore, beyond reproducibility, these results further support the practical importance of our approach: a data-centric method can enable a relatively small open model to achieve performance close to much larger closed-source systems on Isabelle theorem proving.

---

> > ### Author Rebuttal · Reviewer_GjYa · 2026-04-03
> >
> > I appreciate the authors' response, but the rebuttal does not sufficiently address my core concerns regarding the credibility and methodology of the work.
> >
> > First, the provided trace for `amc12b_2020_p21` does not represent an IMO-level problem. Given that the framework reportedly achieves 90.6% on miniF2F-test, it must solve high-difficulty Olympiad problems that require deep, multi-step reasoning. The current rebuttal lacks a detailed trace for a problem at that specific level, making it difficult to verify if the model is performing genuine mathematical reasoning or relying on simpler pattern matching.
> >
> > Second, the supplementary experiments comparing the model to proprietary systems like GPT and Gemini do not reflect the "agentic workflow" I requested. Modern benchmarks for theorem proving agents, such as Hilbert or Numina-lean-agent, involve complex, multi-turn tool interactions and search strategies. Simply reporting Pass@k scores for frontier models does not provide a meaningful baseline for how your data-centric SFT approach compares to state-of-the-art agentic reasoning.
> >
> > Finally, I remain skeptical of the 90.6% result achieved with only 4k training samples and no reinforcement learning. The gap between this ultra-low-resource Isabelle result and the massive synthetic data and RL requirements seen in the Lean community is too significant to ignore. While I value the authors' commitment to open-sourcing the training set and code, the current evidence is not enough to overcome the doubts regarding such highly positive results. I maintain my original assessment.

---

> > > ### Author Response · Authors · 2026-04-05
> > >
> > > Thank you very much for your guidance and suggestions！

---

### Decision · Program_Chairs · 2026-04-30

**Decision:**

Accept (regular)

**Comment:**

This paper focuses on theorem proving in Isabelle, which has fallen behind than Lean. The reviewers view that paper's framework is able to achieve significant improvement. one concern among reviewers was wehther this generalizes to other theorem provers but I don't think that's a very valuable concern as having a state of the art neural theorem prover in Isabelle  is worth of achievement on it's own. Therefore, I am overriding some of the reservations to enthusiastically recommend the paper for acceptance.